# Non-linear interaction between physical activity and polygenic risk score of body mass index in Danish and Russian populations

**Dmitrii Borisevich**[1]*, **Theresia M. Schnurr**[1], **Line Engelbrechtsen**[1,2],
**Alexander Rakitko**[3], **Lars Ängquist**[1], **Valery Ilinsky**[3], **Mette Aadahl**[4,5],
**Niels Grarup**[1], **Oluf Pedersen**[1], **Thorkild I. A. Sørensen**[1,5], **Torben Hansen**[1]

**1** Novo Nordisk Foundation Center for Basic Metabolic Research, Faculty of Health and Medical Sciences, University of Copenhagen, Copenhagen, Denmark, **2** Department of Gynecology and Obstetrics, Herlev Hospital, Herlev, Denmark, **3** Genotek Ltd., Moscow, Russia, **4** Center for Clinical Research and Prevention, Bispebjerg and Frederiksberg Hospital, Frederiksberg, Denmark, **5** Department of Public Health, Faculty of Health and Medical Sciences, University of Copenhagen, Copenhagen, Denmark

* dmitrii@sund.ku.dk

**Data Availability Statement:** The polygenic risk score is available via Online supplementary, https://bmiprsxpa-pa7qfqmwhq-ew.a.run.app/. The

## Abstract

Body mass index (BMI) is a highly heritable polygenic trait. It is also affected by various environmental and behavioral risk factors. We used a BMI polygenic risk score (PRS) to study the interplay between the genetic and environmental factors defining BMI. First, we generated a BMI PRS that explained more variance than a BMI genetic risk score (GRS), which was using only genome-wide significant BMI-associated variants ($R^2 = 13.1\%$ compared to 6.1%). Second, we analyzed interactions between BMI PRS and seven environmental factors. We found a significant interaction between physical activity and BMI PRS, even when the well-known effect of the *FTO* region was excluded from the PRS, using a small dataset of 6,179 samples. Third, we stratified the study population into two risk groups using BMI PRS. The top 22% of the studied populations were included in a high PRS risk group. Engagement in self-reported physical activity was associated with a 1.66 kg/m$^2$ decrease in BMI in this group, compared to a 0.84 kg/m$^2$ decrease in BMI in the rest of the population. Our results (i) confirm that genetic background strongly affects adult BMI in the general population, (ii) show a non-linear interaction between BMI genetics and physical activity, and (iii) provide a standardized framework for future gene-environment interaction analyses.

## Introduction

Body mass index (BMI) is a complex measure that has been robustly associated with cardiometabolic traits and diseases [1]. BMI is a highly heritable complex trait, with heritability estimated to be between 30–40% [2–5]. Studying how genetic variation affects BMI is important to understand the biology of BMI-related diseases.

Genome-wide association studies (GWAS) have identified a multitude of BMI-associated genetic variants at the genome-wide significance threshold ($p < 5 \times 10^{-8}$). The largest meta-

underlying code is available at GitHub, https://github.com/borisevichdi/bmiprs-code. The underlying raw data comprises only sensitive data that cannot be made publicly available: genetic and biological data of human participants. For Inter99 dataset, consent for publication of raw data was not obtained from the individuals, and the dataset could pose a threat to confidentiality. The restriction is imposed by Regional Scientific Ethics Committee (KA 98 155) and the Danish Data Protection Agency. Data access is provided by the Phenomics Platform of the NNF Center for Basic Metabolic Research, https://cbmr.ku.dk/researchfacilities/phenomics/. Point of contact: CBMR-PhenomicsInfo@sund.ku.dk. For Genotek dataset, the user agreement (available at https://www.genotek.ru) states that disclosure of individual-level genetic information and/or self-reported Information to third parties for research purposes will not occur without explicit consent, and the consent was not obtained from the individuals. Due to the user agreement the individual level cannot be made directly available, and the dataset could pose a threat to confidentiality. Data have to be accessed indirectly via Genotek Ltd, https://www.genotek.ru. Data requests should be sent to the Genotek Ltd at info@genotek.ru.

**Funding:** Data collection in the Inter99 study was supported economically by The Danish Medical Research Council (grant nr. 2028-00-0019 and 09-059174), The Danish Centre for Evaluation and Health Technology Assessment (grant nr. 3126-138-1998 and 263-12-1999 and 0-204-03-74), Novo Nordisk, Copenhagen County (grant nr. 9870006), The Danish Heart Foundation (grant nr. 98-2-5-71-22659 and 00-2-9-F4-22872 and 04-10-B201-A309-22171), The Danish Pharmaceutical Association (grant nr. 53-99 and 58-2003), Augustinus foundation (grant nr. 99-1663), Ib Henriksen foundation, and Becket foundation. Novo Nordisk Foundation Center for Basic Metabolic Research is an independent Research Center, based at the University of Copenhagen, Denmark, and partially funded by an unconditional donation from the Novo Nordisk Foundation (www.cbmr.ku.dk) (NNF18CC0034900). Dmitrii Borisevich is receiving funding from NNF Copenhagen Bioscience PhD Programme (NNF17CC0026760). The funders had no role in study design, data collection and analysis, decision to publish, or preparation of the manuscript. Genotek Ltd provided only financial support in the form of AR and VI salaries and the data for the analysis, but did not have any additional role in the study design, data collection and analysis, decision to publish, or preparation of the manuscript. The

analysis of GWAS so far, including ~700,000 adults, identified 941 genetic variants associated with BMI [6]. From the significant variants, genetic risk scores for BMI (BMI GRS) have been constructed representing the number of BMI-increasing risk alleles, weighted by their respective effect sizes within the discovery GWAS. However, the GRS constructed in the study explained only 6.0% of BMI variance, which is substantially less than the estimated heritability of BMI. This finding represents a marked case of missing heritability. There are many potential reasons for heritability missing from the GWAS findings [7], including non-significant variants with small effect size, rare variants, structural variation, and gene-gene and gene-environment interactions.

One of the substantial reasons for the missing heritability of BMI is that BMI is a polygenic trait. Thus the set of genetic variants identified at the genome-wide significance threshold have limited predictive ability. The genetic susceptibility to BMI is accumulated from numerous genetic variants with individually small to modest effects [8]. Recently, computational algorithms have been developed to derive polygenic risk scores (PRS) that combine all available common genetic variants into a single quantitative measure [9]. Applied to BMI, a BMI PRS was shown to be a better predictor of BMI than a BMI GRS comprised of 141 BMI-associated genetic variants [10], as expected based on the highly polygenic nature of BMI.

Another source of the unexplained variation of BMI stems from gene-environment interactions [11, 12]. An interaction occurs when the biological effect of a genetic variant depends on a risk factor, such as an environmental stimulus or a lifestyle factor [13, 14]. For example, physical activity attenuates the effect of a common SNP rs9939609 within the *FTO* locus on BMI [15, 16], the strongest common SNP known to associate with BMI. However, detection of the interactions driven by individual SNPs is challenging. A meta-analysis of 200,452 adults [12] reported only one additional SNP, rs986732, on top of the known *FTO* locus. Attempts to aggregate genetic background using GRS to increase power to detect gene-environment interactions have been made. Recently, a large study in up to ~360,000 unrelated participants from UK Biobank identified several risk factors–alcohol intake, physical inactivity, socioeconomic status, mental health, and sleeping patterns–that influenced the effect of a BMI GRS comprised of 94 BMI-associated genetic variants on BMI [17].

In the present study, we investigated the interactions of a BMI PRS with environmental and lifestyle risk factors and used the interaction to develop a criterion for stratifying a population into two risk groups. We constructed and validated a BMI PRS. We validated an interaction between the BMI PRS and physical activity, which remained highly significant even when omitting *FTO* variants. Finally, we developed a simple non-linear criterion to translate this interaction into clinical practice and future research. We showed that in a subset of individuals with the highest 22% of BMI PRS values, self-reported physical activity was associated with a two-fold higher difference in BMI than in the remaining 78% of the individuals.

## Materials and methods

### Overall analysis workflow

The workflow of the analysis is present at Fig 1. The polygenic risk score of BMI was built on UK Biobank and GIANT summary statistics as the source dataset and a "Training" subset from the Inter99 cohort as the target dataset, as described in the "BMI Polygenic Risk Score" section. The resulting PRS was validated in an independent "Validation/Discovery" subset of Inter99, using phenotypes described in the "Phenotypes" section. Both parts of the Inter99 dataset are described in the "Inter99 dataset" section. Interaction analyses were done in the same subset of Inter99 between the BMI PRS and risk factors described in the "Risk Factors" section. Both analyses were performed, as described in the "Statistical Analysis" section. A *post*

specific roles of these authors are articulated in the 'author contributions' section.

**Competing interests:** This submission contains original research. Novo Nordisk provided unrestricted grants for data collection. No authors were employed by or consulted Novo Nordisk during this study, and no conflict of interest exists in connection to patents, products in development, marketed products, or alike. This funding does not alter our adherence to PLOS ONE policies on sharing data and materials, and does not impose restrictions on sharing of data and/or material. AR and VI are employees of Genotek Ltd. and may own stock/stock options in the company. No other conflict of interest exists in connection to patents, products in development, marketed products, or alike. This funding does not alter our adherence to PLOS ONE policies on sharing data and materials, and does not impose restrictions on sharing of data and/or material. Other authors declare no conflict of interests.

*hoc* criterion to stratify the individuals was developed using the "Validation/Discovery" subset, as described in the "Building the PRS Criterion" section. The criterion was validated in an independent "Replication" Genotek cohort of different origin, described in the "Genotek Dataset" section.

## BMI polygenic risk score

To generate a polygenic risk score (PRS), a source summary statistics dataset from the relevant GWAS study and a target cohort dataset with individual genotypes and phenotypes are required. A meta-analysis of UK Biobank and GIANT [6] (N ~ 700,000) was used as the source of summary statistics. PRS was generated using *LDpred* tool (v.1.0.6) [9] and its standard workflow. Briefly, a Danish population-based cohort Inter99 [18] (N = 6,179) was used. The cohort was randomly split into training (N = 1,000) and validation (N = 5,179) subsets. SNPs available in the input data, comprising subsets and the summary statistics of the BMI GWAS, were aligned using *LDpred coord* command. Eleven BMI PRS were generated for the different parameter *f*, representing what *LDpred* calls the "fractions of causative variants". The following parameters *f* were used to cover different orders of magnitude: 1.0, 0.3, 0.1, 0.03, 0.01, 0.003, 0.001, 0.0003, 0.0001, $3 \times 10^{-5}$, $1 \times 10^{-5}$. The scores were generated using *LDpred gibbs* and their performance was assessed in the training subset only using $R^2$ provided by *LDpred score* command. The best performing score was selected. The score values were calculated for the samples in the validation subset using *LDpred score* command. These values and this subset were used for the subsequent analyses.

A genetic risk score (GRS) was constructed using only the genome-wide significant (p < 5 $\times 10^{-8}$) variants from the same meta-analysis of UK Biobank and GIANT [6] and calculated in the validation subset of the Inter99 dataset.

To exclude the *FTO* effect from the PRS, all the SNPs within ± 100,000 nucleotides from the lead SNP rs9939609 were excluded from the score. This covered at least all the SNPs with estimated linkage disequilibrium $R^2 \geq 0.1$.

## Phenotypes

We have reviewed the literature and created a shortlist of cardiometabolic phenotypes, which were previously reported to associate with BMI. For evaluation of the association between the BMI PRS and cardiometabolic traits, we selected well-known obesity-related traits. The literature review highlighted multiple biochemical and anthropometric measurements and functional tests associated with BMI. The following twenty phenotypes were available in the Inter99 cohort for analysis:

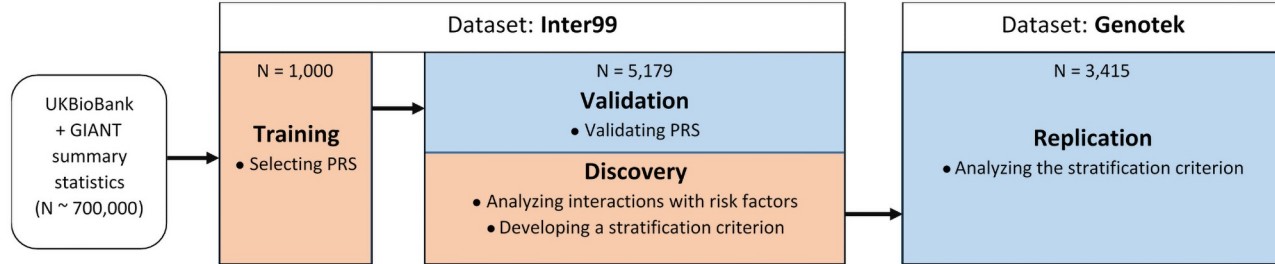

**Fig 1. The overall workflow of the analysis.** The flow of the data and the steps of the analysis. The subsets of data are named "Training", "Validation/ Discovery", and "Replication", according to the naming in the text. Orange color highlights steps of analysis where models were trained, blue color highlights steps of analysis where models were validated. The N = 5,179 subset of Inter99 was used both for the validation of the BMI PRS and for the discovery of the interactions with risk factors.

- Dyslipidemia markers: fasting total cholesterol, high-density lipocholesterol, triglycerides,

- Inflammation markers: fasting serum C-reactive protein (high sensitive hs-CRP), interleukin-18 (IL18),

- Cardiovascular diseases markers: diastolic blood pressure, systolic blood pressure, pulse pressure,

- Glucose level markers: hemoglobin A1C, fasting plasma glucose, 30 min & 120 min plasma glucose during oral glucose tolerance test (OGTT),

- Insulin sensitivity or resistance markers: homeostatic model assessment of insulin resistance (HOMA-IR), fasting serum insulin,

- Satiety markers: fasting leptin (LEP),

- Anthropometrics: hip, waist circumference, waist-to-hip ratio, weight, height.

### Risk factors

Different behavioral and environmental traits (called together "risk factors" in the text for simplicity) were shown to affect the BMI via a gene-environment interaction [17] using a GRS approach. While Inter99 did not contain exactly the same risk factors, we have found similar risk factors available in Inter99. The following factors were matched to the previously reported and analyzed in this study:

- Smoking status [19], four categories: "smoking daily", "smoking occasionally", "never smoking", and "previously smoking" (N = 5,155).

- Alcohol consumption [20], two categories: "no or moderate alcohol consumption" defined as $\leq$ 6 units/week for women and $\leq$ 12 units/week for men; "high alcohol consumption" defined as > 6 units week for women and > 12 units/week for men. 1 unit is equivalent to 12 g of pure alcohol (N = 5,002).

- Diet quality, two categories: "poor diet" defined as 4–8 points on the diet quality score (DQS) system described at [21]; "healthy diet" defined as $\geq$ 9 points on the DQS score system (N = 5,020).

- Physical activity level [22], two categories: "inactive" defined as self-reported commuting and leisure-time physical activity $\leq$ 225 min/week; "active" defined as self-reported commuting and leisure-time physical activity > 225 min/week (N = 4,859).

- Mental health [23], two categories: "high" defined as mental health component score (MCS) higher than the 75th percentile within the study population of the same sex; "low" defined as MCS lower than the 75th percentile. MCS has been calculated as described in [23], using the Short Form 12 (SF-12) questionnaire [24] (N = 4,878).

- Quality of sleep, two categories: "good" defined by the answer 'no' to the question 'do you often suffer from insomnia'; "poor" defined by the answer 'yes' (N = 5,129).

- Socioeconomic class [20], five categories: "not working, no education", "not working, $\geq$ 1 year of education", "working, no education", "working, 1–3 years of education", "working, $\geq$ 4 years of education". Education is counted after mandatory school years. The categories were combined from education and employment statuses, reported in [20] (N = 4,807).

All risk factors were measured by self-report questionnaire in the Inter99 cohort.

## Statistical analysis

The validation subset of the Inter99 was used for the analysis of associations. Exploratory data analyses and linear regression analyses were done using *python* (Python 3, including *numpy*, *pandas*, *matplotlib*, *seaborn* and *statsmodels* packages).

Explained variances of the BMI by the PRS and GRS were calculated using ordinary least squares models. For regression analysis, robust linear models were used. Formulas are provided inline in the Results text using the *python statsmodels* standard, identical to the R *glm* standard [25]. Terms with "*C()*" represent categorical phenotypes. Terms, combined using the semicolon as "*X:Y*", represent the interaction between *X* and *Y*. Term "*1*" represents Intersect.

Findings were reported if they have passed a study-wise Bonferroni-adjusted p-value significance cut-off. I.e., the Bonferroni correction for multiple testing was applied for each analysis separately. Specifically, p-values in the analysis of associations of cardiometabolic phenotypes with the BMI PRS were adjusted for the number of the outcome phenotypes tested (twenty tests, $p = 2.5 \times 10^{-3}$). Analyses for the associations and interactions with categorical risk factors were each independently adjusted for the number of associations or interactions tests performed (twelve tests, $p = 4.17 \times 10^{-3}$).

The presence of the non-linear associations between the BMI PRS and the BMI was analyzed by regressing out the linear effect of the PRS and covariates (age and sex) and checking if the square of the PRS was significantly associated with the residuals.

## Building the PRS criterion

The selection of the PRS cut-off for the stratification was made in the discovery subset of the Inter99. Ordinary least squares model "*BMI ~ 1 + age + C(sex)*" was used to regress out intercept, age, and sex fixed effects. Residuals of BMI were fitted to a family of models of the form "*BMI residuals ~ 0 + PRS + C(physical activity) + C(physical activity):I(PRS > $i^{th}$ percentile)*". "*I (PRS > $i^{th}$ percentile)*" is a binary indicator, *true* when *PRS* is greater than the $i^{th}$ percentile of observed PRS values and *false* otherwise. One hundred models with cut-offs *i = 0,1,..,99* were calculated. The cut-off *i* producing a model with the highest $R^2$ was selected.

The criterion model was combined as "*BMI ~ 1 + age + C(sex) + PRS + C(physical activity) + C(physical activity):I(PRS > $i^{th}$ percentile)*". It splits all the population individuals into two groups based on their BMI PRS for measuring the interaction with physical activity, instead of using precise PRS values. The model was validated and analyzed in the independent Genotek replication dataset.

## Inter99 dataset

Inter99 is a previously described Danish population-based dataset consisting of 6,179 individuals [18]. The study was approved by the Regional Scientific Ethics Committee (KA 98 155) and the Danish Data Protection Agency.

Briefly, all individuals were genotyped using Illumina HumanOmniExpress-24 (v1.0A / v1.1A). Genotypes were called using the Genotyping module (version 1.9.4) of GenomeStudio software (version 2011.1; Illumina). Only individuals having a call rate $\geq$98% were included. Monomorphic SNPs and SNPs with Hardy-Weinberg expectation p-value $< 10^{-5}$ were excluded. Genotypes were imputed using the Michigan imputation server with the HRC1.1 [26] reference panel.

The dataset was randomly split into two subsets. A "Training" subset (N = 1,000) was used for PRS generation. A "Validation/Discovery" subset (N = 5,179) was used for the analysis of the PRS, its interactions with risk factors, and building the stratification criterion.

### Genotek dataset

Genotek is an unpublished Russian population-based set consisting of 3,415 unrelated individuals aged between 20 and 60 years old with self-reported measures.

All individuals were genotyped using Illumina Infinium Global Screening Array (v1.0 / v2.0). SNPs with call rate <0.9, the calls on Y chromosome for women and the heterozygous calls on X chromosome for men were removed. Genotypes were imputed using BEAGLE 5.1 with the HRC reference panel. After imputation, variants with MAF below 1% or $DR^2$ below 0.7 were excluded.

The following phenotypes were used from questionnaire information: age, weight, height, sex, and physical activity. Three categories of physical activity levels were available: "sedentary" defined as self-reported "I have sedentary lifestyle", "moderate" defined as self-reported "I take walks every single day", and "high" defined as self-reported "My job involves physical activity, or I do a lot of sports". To match the physical activity measures between the datasets, the "sedentary" group from the Genotek dataset was considered equivalent to the "inactive" group from the Inter99 dataset, and the "moderate" and "high" groups from the Genotek dataset were considered equivalent to the "active" group from the Inter99 dataset.

The dataset was used for validation of the stratification criterion.

## Results

### BMI polygenic risk score

To have a tool for measuring the genetic susceptibility to BMI, we have constructed a BMI PRS. BMI PRS was a score corresponding to the *LDpred* parameter "fraction of causative variants" *f* = 0.3, selected using a procedure described in "Materials and methods". The explained variance of the score was 13.0% in a training subset (N = 1,000) and 13.1% in a validation subset (N = 5,179). In comparison, the best currently available GRS of BMI, which utilizes 941 SNPs [6], explained only 6.1% of the variance in the same validation subset. In the validation subset, we observed a significant association between the BMI and the BMI PRS (p = 3.36 x 10$^{-172}$, linear regression *BMI ~ 1 + PRS*). The difference in the median BMI between the top and the bottom deciles of the individuals, according to their PRS, was 5.17 kg/m$^2$ (Fig 2A), also showing an improvement over the GRS, which showed a median difference of 3.41 kg/m$^2$ (Fig 2B) in the same dataset.

To check if the BMI PRS captures other BMI-related risk phenotypes, we analyzed whether known BMI-associated cardiometabolic traits were also correlated with BMI PRS. First, we confirmed that BMI was associated with all twenty known cardiometabolic traits in our dataset (p < 2.5 x 10$^{-3}$) (S1 Table). Next, we analyzed associations between the BMI PRS and the cardiometabolic traits. Each of the cardiometabolic traits was also associated (p < 2.5 x 10$^{-3}$) with the generated BMI PRS (S1 Table), when adjusted for age, sex and genetic principal components (gPCs) (linear regression *trait ~ 1 + PRS + age + C(sex) + gPC1 + gPC2 + gPC3)*. The distributions of all traits across individuals stratified by PRS deciles are shown in S1 File.

While GWAS is based on univariate linear associations, the cumulative increase in risk may be non-linear. To understand if the impact of the genetic load for BMI was linear, we checked for the non-linear effects of the BMI PRS. We observed a significant association between the square of BMI PRS and the BMI residuals corrected for the covariates (age and sex) and the

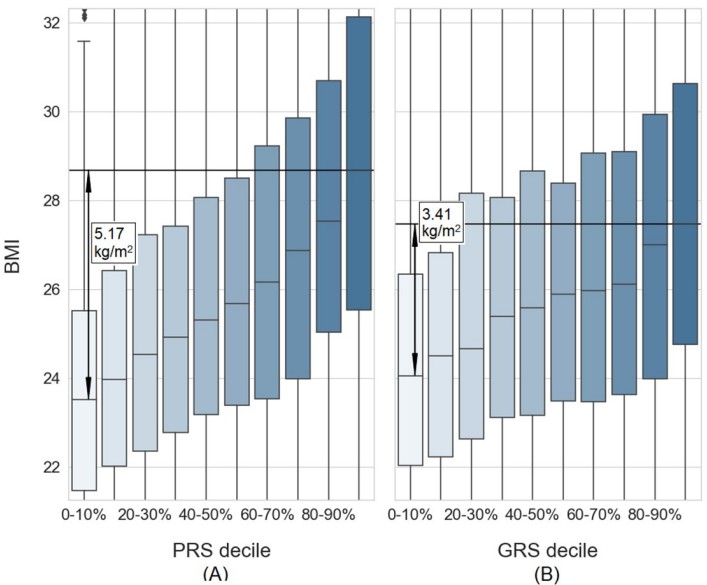

**Fig 2. BMI distribution of individuals, stratified by genetic risk.** (A): Individuals stratified by BMI PRS deciles. (B): Individuals stratified by BMI GRS deciles. Both graphs are plotted in the same scale and cut at 10th / 90th percentiles on the y-axis. Box plots represent the median and 25th / 75th percentiles. Whiskers are set at 1.5 IQR. Both graphs show BMI, unadjusted for age and sex. Adjusted BMI residuals and outliers are available at Online Supplementary.

**Table 1. Associations of the risk factors with BMI.**

| Risk factor | Category * | Effect size | P-value |
|---|---|---|---|
| Smoking | Never smoker | Reference group | |
| | Previous smoker | + 0.06 kg/m$^2$ | 0.648 |
| | Smoking occasionally | + 0.27 kg/m$^2$ | 0.341 |
| | Smoking daily | - 1.06 kg/m$^2$ | $6.49 \times 10^{-18}$ |
| Physical activity | Active | Reference group | |
| | Inactive | + 0.79 kg/m$^2$ | $8.49 \times 10^{-13}$ |
| Alcohol consumption | No or moderate | Reference group | |
| | High | - 0.36 kg/m$^2$ | $9.00 \times 10^{-4}$ |
| Diet quality | Healthy | Reference group | |
| | Poor | - 0.25 kg/m$^2$ | 0.024 |
| Socioeconomic class | Working, $\geq$ 4 years of education | Reference group | |
| | Working, 1–3 years of education | + 0.00 kg/m$^2$ | 0.99 |
| | Working, no education | + 0.21 kg/m$^2$ | 0.24 |
| | Not working, $\geq$ 1 year of education | + 0.18 kg/m$^2$ | 0.40 |
| | Not working, no education | + 0.01 kg/m$^2$ | 0.98 |
| Mental health | High | Reference group | |
| | Low | - 0.06 kg/m$^2$ | 0.62 |
| Quality of sleep | Good | Reference group | |
| | Poor | + 0.15 kg/m$^2$ | 0.28 |

* The categories are described in the "Risk Factors" section of Materials and methods.

Effect sizes and p-values for each category are reported relative to the Reference group.

BMI PRS ($p = 2.61 \times 10^{-3}$). In contrast, BMI GRS did not show significant non-linear effects ($p = 0.50$).

No evidence was observed for associations between BMI and any of the three genetic principal components (linear regression *BMI ~ 1 + PRS + age + C(sex) + gPC1 + gPC2 + gPC3*, $p > 0.05$). To simplify the analysis, we have excluded the adjustment for the gPCs in the subsequent analysis.

## Associations between the BMI-associated risk factors and BMI

Several risk factors (environmental stimuli and lifestyle factors) are known to be associated with a change in BMI as described in Materials and methods. To study these factors, we first validated associations between them and BMI in our dataset. BMI was associated (Table 1) with smoking, physical activity, and alcohol consumption groups (twelve tests performed, $p < 4.17 \times 10^{-3}$) when adjusted for age, sex, and BMI PRS (linear regression *BMI ~ 1 + PRS + age + C(sex) + C(risk factor)*). The same risk factors were found to be significantly associated when using the BMI GRS instead of the BMI PRS in our data (S2 Table). BMI was only nominally ($p < 0.05$) associated with diet quality, and BMI was not associated with socioeconomic class, mental health, or quality of sleep in our dataset (Table 1).

## Interactions between the BMI PRS and BMI-associated risk factors

To analyze if the presence of the risk factor alters the effect of BMI PRS on BMI, we examined potential interactions between the BMI PRS and all seven risk factors (linear regression *BMI ~ 1 + PRS + age + C(sex) + C(risk factor) + PRS:C(risk factor)*). We found a significant interaction (twelve tests performed, $p < 4.17 \times 10^{-3}$) only between the BMI PRS and physical activity. Physically active individuals demonstrated 0.81 kg/m$^2$ ($p = 2.24 \times 10^{-13}$) lower BMI than inactive and an additional 0.33 kg/m$^2$ ($p = 3.13 \times 10^{-3}$) lower BMI per each standardized BMI PRS unit. Unadjusted BMI values per BMI PRS decile are visualized in Fig 3A. Unlike the generated BMI PRS, the BMI GRS did not demonstrate significant ($p < 4.17 \times 10^{-3}$) interactions with any of the risk factors, but it showed a nominal significance ($p = 0.045$) for the interaction with physical activity in the same direction as PRS. Unadjusted BMI values per BMI GRS decile are visualized in Fig 3B. Physical activity was not itself significantly associated with the BMI PRS (logistic regression *physical activity ~ 1 + age + C(sex) + PRS*, $p = 0.053$).

The *FTO* locus is the primary locus known to interact with physical activity. In particular, the minor allele of the *FTO* lead SNP rs9939609 is associated with increased BMI. This effect was shown to be attenuated in physically active individuals in Inter99 before [15]. To check if the BMI PRS interaction with physical activity was driven only by the effect of *FTO*, we have performed the interaction analysis with a PRS without the *FTO* region. We still demonstrated a significant interaction between the residual PRS and physical activity ($p = 3.23 \times 10^{-3}$).

The observed BMI PRS interaction with physical activity was reproduced in a replication dataset of different origin (Russian). In this dataset, physically active individuals had 0.97 kg/m$^2$ ($p = 9.10 \times 10^{-14}$) lower BMI than inactive individuals and an additional 0.45 kg/m$^2$ ($p = 5.57 \times 10^{-4}$) lower BMI per one standardized BMI PRS unit.

## A criterion for stratification based on the BMI PRS

In clinical genetics practice, simple binary criteria are preferred over numerical variables. To utilize our findings in future studies, we have in *post hoc* analysis developed a PRS-based criterion to divide the individuals into two different groups of genetic risk in a clinical setting. We have selected the cut-off between the groups in a data-driven manner by selecting the highest $R^2$ for BMI (S1 Fig, Materials and methods). The cut-off was selected at PRS = 78$^{th}$ percentile

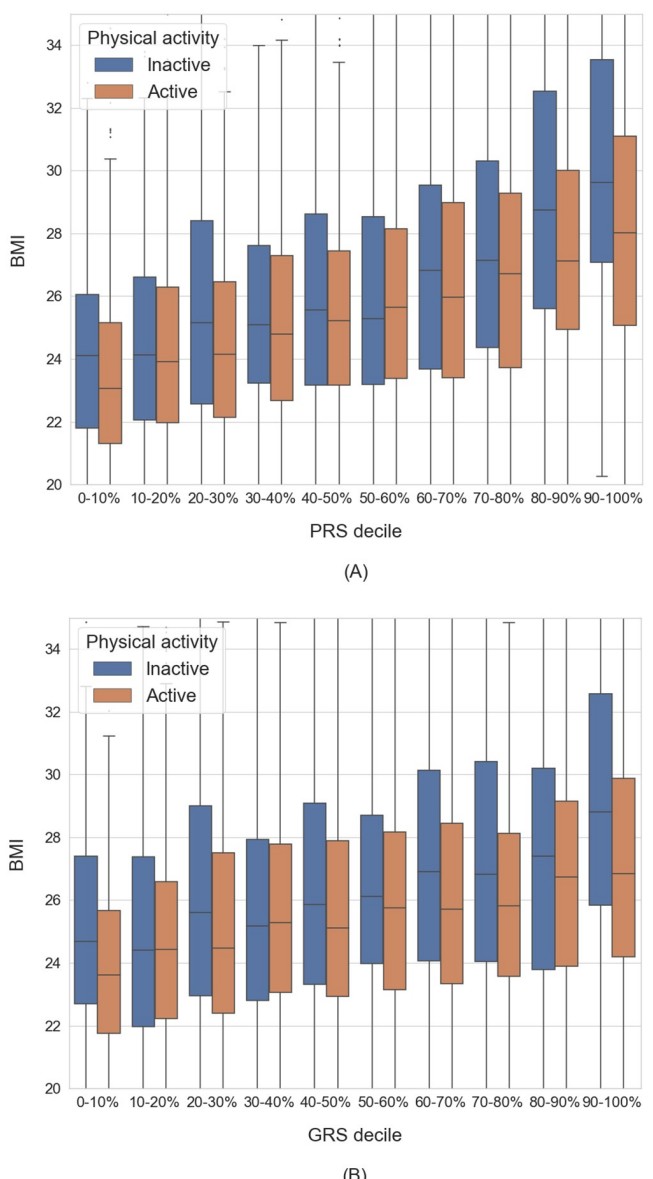

**Fig 3. BMI distribution of individuals, stratified by genetic risk and levels of physical activity.** (A): Individuals stratified by BMI PRS deciles. (B): Individuals stratified by BMI GRS deciles. Both graphs are plotted in the same scale and cut at $10^{th}$ / $90^{th}$ percentiles on the y-axis. Box plots represent the median and $25^{th}$ / $75^{th}$ percentiles. Whiskers are set at 1.5 IQR. Both graphs show BMI, unadjusted for age and sex. Adjusted BMI residuals and outliers are available at Online Supplementary.

(PRS = 0.783 units in our dataset), dividing all the individuals into two groups. Here we coin the term "PRS$_{0-78\%}$" group for the individuals with PRS < $78^{th}$ percentile, and the "PRS$_{78-100\%}$" group for the individuals with PRS ≥ $78^{th}$ percentile. We observed a significant interaction between the two risk groups of BMI PRS and physical activity (p = 2.83 x $10^{-6}$, regression model *BMI ~ 1 + age + C(sex) + PRS + C(exercise)* I(PRS > $78^{th}$ percentile)). In the PRS$_{0-78\%}$ group, the average BMI of the inactive individuals was only 0.65 kg/m$^2$ higher than the average BMI of the active individuals. In the PRS$_{78-100\%}$ group, the average BMI of the inactive

individuals was 2.07 kg/m$^2$ higher than the average BMI of the active individuals. The BMI distribution is shown in Fig 4A.

We replicated the interaction between BMI PRS risk groups and physical activity at the 78% cut-off in the replication dataset. In the PRS$_{0-78\%}$ group, the average BMI of the inactive individuals was 0.84 kg/m$^2$ higher than the average BMI of the active individuals. In the PRS$_{78-100\%}$ group, the average BMI of the inactive individuals was 1.66 kg/m$^2$ higher than the average

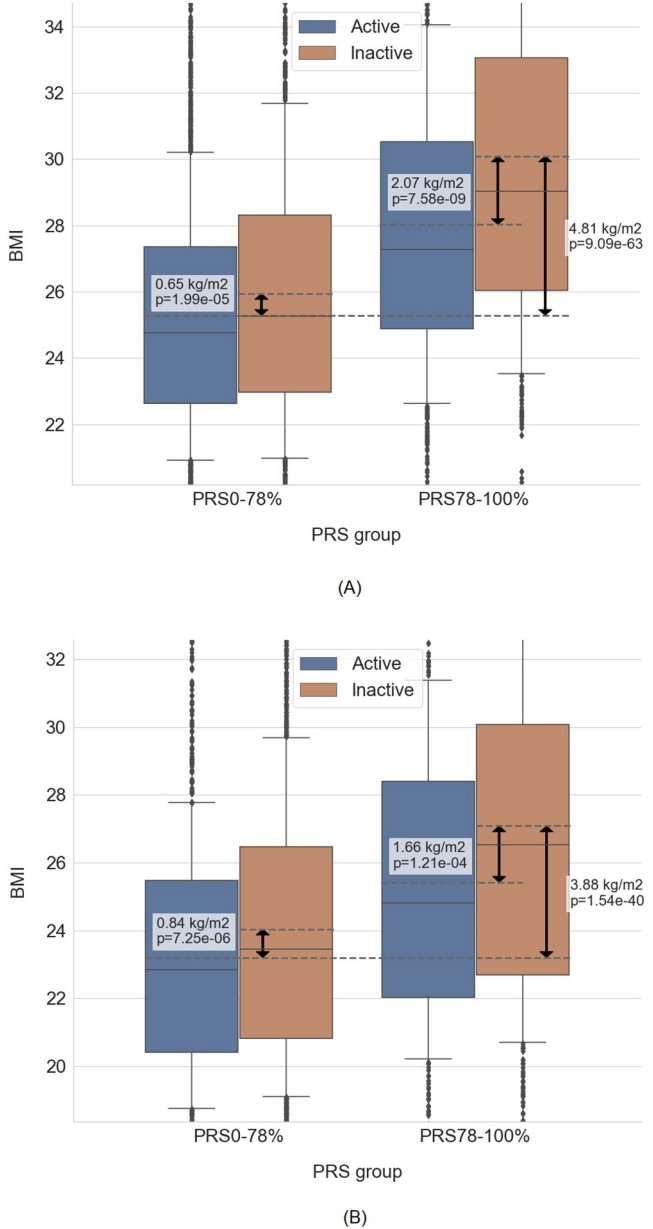

**Fig 4. BMI distribution of individuals, stratified by PRS groups and levels of physical activity.** (A): Discovery Inter99 dataset. (B): Validation in Genotek dataset. Graphs are cut at 10% / 90% quantiles on the y-axis (note the different scales on the panels). Solid lines represent median, boxplot outlines represent 25% / 75% quantiles, whiskers represent 10% / 90% quantiles. Dashed lines represent the mean of the respective box, which were compared in the statistical test. The difference in mean values and p-value of respective tests are shown on the plot. Both graphs show BMI, unadjusted for age and sex. Adjusted BMI residuals and outliers are available at Online Supplementary.

BMI of the active individuals. The BMI distribution is shown in Fig 4B. The addition of the cut-off interaction increased $R^2$ by +0.55% compared to the model without interaction. In comparison, the linear interaction increased $R^2$ only by +0.22%.

## Discussion

In this study, we have detected a non-linear interaction between BMI genetics and physical activity using BMI PRS. We have constructed a BMI PRS and assessed its performance. The BMI PRS demonstrated a substantial improvement in the explained variance of BMI over the BMI GRS. Using the BMI PRS, we have detected an interaction between the genetic component of BMI and physical activity in a relatively small dataset. This interaction was neither limited to the interaction driven by the *FTO* locus nor significant when using the BMI GRS for the same analysis. The application of PRS enabled us to identify 22% of individuals with the highest PRS. In this group, self-reported physical activity was associated with a 2-fold higher difference in BMI than in the remaining 78% of study participants. The model with a non-linear two-group division of individuals showed higher $R^2$ than a model with a linear interaction.

Studies of BMI indicate that BMI is a highly heritable phenotype. However, existing tools to measure the genetic predisposition to BMI, namely BMI GRS, fail to capture most of this heritability. As a result, the predictive power of BMI GRS is limited, preventing BMI GRS from being used in clinical practice. Lack of proper genetic tools to assess predisposition to BMI hinders progress in this area of research. To address this limitation we used a polygenic risk scoring approach. Studying interactions between the environmental and behavioral risk factors and BMI genetics is important to understand the mechanisms by which risk factors modify genetic predisposition to BMI. To address this we analyze interactions between BMI PRS and seven previously known risk factors.

Different socioeconomic and behavioral factors are known to affect BMI. Here we demonstrate a particularly strong connection between the genetics of BMI and physical activity. Physical activity levels have been shown to interact with BMI-associated variants using the GRS approach [17]. Here we also demonstrate the interaction between the BMI PRS and physical activity levels. Interestingly, this interaction was not significant when applying the BMI GRS, although we replicated the interaction effect in the same direction with nominal significance. This probably means that our study sample size was not large enough for the GRS to show a significant association. Application of the BMI PRS, whose interaction with physical activity was stronger than that of the BMI GRS, likely enabled us to see this interaction in a smaller dataset (N = 5,179) than the one used before (N = 362,496) [17]. It supports the idea that BMI PRS may enable analyses of phenotypes and their interactions in small datasets. This property of PRS would allow detailed phenotypes that are difficult to sample and are lacking from large biobanks to be studied using a PRS approach.

The SNP rs9939609 (also commonly referred to by the nearby gene name, *FTO*) is of particular interest for studies of obesity. Previously, the effect of *FTO* locus on BMI has been shown to interact with physical activity in a European population analysis [12]. When we excluded the *FTO* locus from the BMI PRS, the remaining PRS still showed significant interaction with physical activity, in line with previous GRS studies [17]. This observation indicates that our PRS captures more than just an interaction of physical activity with *FTO* by integrating weaker interactions of other SNPs. This demonstrates that a PRS may facilitate analyses of the interaction between BMI-associated genetic markers and physical activity in future studies.

We succeeded in replicating the interaction between physical activity and the BMI PRS observed in the Danish population. We replicated it in a Russian population sample with a similar effect size, despite genetic and cultural differences between the two populations. PRS

prediction accuracy is known to decline when applied in populations different from the origin population of the source summary statistics [27]. In this study, we utilized previously unpublished data from a Russian study sample. Despite being geographically close, genetic PCA [28] clearly separates the Russian from the European population from which the GWAS was derived. Successful replication in Russians strengthens our findings, suggesting that both our European-based BMI PRS and the observed cut-off are generalizable and may be applied in other populations.

One issue we observed with the replication dataset was that $R^2$, explained by the regression models, was higher on average in the replication dataset than in the discovery dataset. This difference was driven by higher $R^2$ explained by age and sex alone. $R^2$ for the model *BMI ~ 1 + C (sex) + age* was 21.5% in the replication dataset, while only 2.7% in the discovery dataset. The individuals' age, height, and weight in the replication dataset were self-reported, in contrast to the discovery dataset, where these measures were collected objectively. We speculate that the observed difference in the $R^2$ might have been caused by biased reporting of variables by individuals in the replication sample.

There are few potential caveats in the performed research. First, the observed interaction between BMI PRS and physical activity should be interpreted cautiously. The association between weight and physical activity is believed to be bi-directional [29]. Reduction in physical activity directly shifts energy balance in favor of weight gain. However, increased BMI also causes a reduction in physical activity, which may create artifacts in the analysis of the interaction between the BMI PRS and physical activity. Our study is cross-sectional, and it cannot provide insights into directionality nor causality. In our cohort, 60–70% of the individuals within any BMI PRS decile were physically active, and physical activity did not decrease significantly with increased BMI PRS. Therefore, we speculate that the observed interaction is caused by physical inactivity directly and is not an artifact. If this hypothesis is correct, our findings would indicate that physical activity would be more beneficial for individuals with a genetic predisposition to high BMI than for individuals without such predisposition.

Second, the sample size of our study (N = 5,179) was relatively small for reaching high levels of significance in performed tests and drawing strong conclusions. To resolve this issue, we have used study-wise correction for multiple testing. I.e., each analysis was corrected for multiple testing in the number of phenotypes independently, as described in detail in Materials and methods. Stronger claims could be made in future studies, where more samples would be available.

Third, in this paper we focus on BMI, but there is a piling amount of evidence that other body composition measures are important, when assessing obesity. Fat vs. fat free mass distribution is an important measure [30], capturing differences in body constitution between the people with the same BMI. Another important set of measures are waist-to-hip ratio, waist circumference and similar [31], capturing the fat distribution around the body. These measures are orthogonal to BMI. Future studies of genetics of these parameters using PRS approach could provide additional value to understanding of obesity.

Last, typically, the cut-offs for "high PRS" groups are selected arbitrarily by picking PRS percentiles [32] or selecting a group based on the corresponding increase in risk [33]. We decided not to select the BMI PRS cut-off for population stratification *a priori* but to create it as a part of a *post hoc* analysis. In our study, we have optimized the cut-off to increase the variance explained by the resulting criterion. Such a criterion would perform better than an arbitrary selected one, but it would also be more prone to overfitting. To ensure that the resulting cut-off is not an artifact of overfitting, we have performed an independent replication. We found the cut-off using the Inter99 dataset as a discovery dataset and then validated and analyzed the criterion in an independent Genotek dataset. The replication was successful, and we

observed similar effect sizes and improvements in the regression models' performance. These observations prove the criterion validity and generalizability.

Application-wise, BMI PRS may support a path towards personalized obesity prevention and treatment. BMI PRS enabled us to make a criterion to stratify the population into two groups–$PRS_{0-78\%}$ and $PRS_{78-100\%}$, with genetic predispositions to different BMI levels. These two groups showed a difference in the sizes of the physical activity effects on BMI. Using the criterion, we have observed that for 78% of the examined population sample, the average increase in BMI associated with physical inactivity was 0.84 kg/m$^2$. In comparison, for the top 22%, the average increase was 1.66 kg/m$^2$. This cut-off interaction model also provided a larger explained variance than the linear interaction model. Together, the large effect size observed only in the $PRS_{78-100\%}$ group, and the simplicity of the criterion, open a path for better research and clinical practice. The criterion may be applied to future recall-by-genotype intervention to dissect the causal interplay between the BMI-associated genetic factors and physical activity in defining BMI. If physical inactivity is shown to drive the BMI increase, while BMI PRS defines how large the increase will be, then differential prevention of obesity could be implemented based on the individual's genetic risk group.

In conclusion, our work showcases an important example of enabling gene-environment interactions analyses by using PRS as a genetic tool and provides a useful criterion for future genetic studies of BMI. Our results demonstrate that a BMI PRS is a better instrument to measure the genetic predisposition to BMI than a BMI GRS. By discovering the interaction between BMI PRS and physical activity, we show how a PRS may be used for studying gene-environment interactions in small datasets, where a GRS is unlikely to reveal significant findings. The developed cut-off model shows how findings from analyzing a PRS may be translated into clinical research. While the interaction between the BMI PRS and physical activity warrants a careful interpretation, we suggest that our work may support the path towards personalized physical activity-based prevention of obesity based on genetic risk.

## Supporting information

**S1 File. Distributions of the BMI-associated phenotypes of individuals, stratified by BMI polygenic risk score (PRS) deciles.**
(ZIP)

**S1 Fig. Performance of models using two groups of genetic risk for interaction with physical activity.**
(DOCX)

**S1 Table. p-values of associations between cardiometabolic traits and BMI or BMI PRS.**
(XLSX)

**S2 Table. Associations of the risk factors with BMI adjusted for BMI GRS.**
(XLSX)

**S1 Appendix. List of online supporting materials.**
(DOCX)

## Acknowledgments

The authors would like to thank Sophia Metz and Jonathan J. Thompson for an independent review of the manuscript and valuable feedback.

## Author Contributions

**Conceptualization:** Dmitrii Borisevich, Theresia M. Schnurr, Line Engelbrechtsen, Alexander Rakitko, Lars Ängquist, Thorkild I. A. Sørensen, Torben Hansen.

**Data curation:** Dmitrii Borisevich, Theresia M. Schnurr, Line Engelbrechtsen, Alexander Rakitko, Valery Ilinsky, Mette Aadahl, Niels Grarup, Oluf Pedersen, Torben Hansen.

**Formal analysis:** Dmitrii Borisevich, Alexander Rakitko.

**Funding acquisition:** Niels Grarup, Oluf Pedersen, Torben Hansen.

**Investigation:** Dmitrii Borisevich, Theresia M. Schnurr, Line Engelbrechtsen, Alexander Rakitko, Mette Aadahl.

**Project administration:** Dmitrii Borisevich, Valery Ilinsky, Torben Hansen.

**Resources:** Valery Ilinsky, Mette Aadahl, Niels Grarup, Oluf Pedersen.

**Software:** Dmitrii Borisevich, Alexander Rakitko, Lars Ängquist.

**Supervision:** Valery Ilinsky, Niels Grarup, Oluf Pedersen, Torben Hansen.

**Visualization:** Dmitrii Borisevich, Alexander Rakitko.

**Writing – original draft:** Dmitrii Borisevich, Theresia M. Schnurr, Line Engelbrechtsen, Thorkild I. A. Sørensen.

**Writing – review & editing:** Dmitrii Borisevich, Theresia M. Schnurr, Line Engelbrechtsen, Alexander Rakitko, Lars Ängquist, Valery Ilinsky, Mette Aadahl, Niels Grarup, Oluf Pedersen, Thorkild I. A. Sørensen, Torben Hansen.

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
