## [Decision Letter · Decision Letter 0]

28 Jun 2021

PONE-D-21-17351

Non-linear interaction between physical activity and polygenic risk score of body mass index in Danish and Russian populations

PLOS ONE

Dear Dr. Borisevich,

Thank you for submitting your manuscript to PLOS ONE. After careful consideration, we feel that it has merit but does not fully meet PLOS ONE’s publication criteria as it currently stands. Therefore, we invite you to submit a revised version of the manuscript that addresses the points raised during the review process.

We look forward to receiving your revised manuscript.

Kind regards,

David Meyre

Academic Editor

PLOS ONE

Journal Requirements:

3. PLOS journals require authors to make all data underlying the findings described in their manuscript fully available without restriction unless the data are subject to ethical restrictions or owned by someone other than the authors (https://journals.plos.org/plosone/s/data-availability#loc-acceptable-data-access-restrictions). Therefore, we ask that you please upload underlying data to an appropriate data repository and update your Data Availability Statement accordingly or provide all contact details for where an interested researcher would need to apply to gain access to the relevant data. Please note that it is not acceptable for an author to be the sole named individual responsible for ensuring data access.

4. Thank you for stating the following in the Financial Disclosure section:

"Data collection in the Inter99 study was supported economically by The Danish Medical Research Council (grant nr. 2028-00-0019 and 09-059174), The Danish Centre for Evaluation and Health Technology Assessment (grant nr. 3126-138-1998 and 263-12-1999 and 0-204-03-74), Novo Nordisk, Copenhagen County (grant nr. 9870006), The Danish Heart Foundation (grant nr. 98-2-5-71-22659 and 00-2-9-F4-22872 and 04-10-B201-A309-22171), The Danish Pharmaceutical Association (grant nr. 53-99 and 58-2003), Augustinus foundation (grant nr. 99-1663), Ib Henriksen foundation, and Becket foundation. Novo Nordisk Foundation Center for Basic Metabolic Research is an independent Research Center, based at the University of Copenhagen, Denmark, and partially funded by an unconditional donation from the Novo Nordisk Foundation (www.cbmr.ku.dk) (NNF18CC0034900). Dmitrii Borisevich is receiving funding from NNF Copenhagen Bioscience PhD Programme (NNF17CC0026760).

We note that you received funding from a commercial source: Novo Nordisk

"I have read the journal's policy and the authors of this manuscript have the following competing interests: Alexander Rakitko and Valery Ilinsky are employees of Genotek Ltd. The rest of the authors declare no competing interests."

We note that one or more of the authors are employed by a commercial company: Genotek Ltd.

5.1. Please provide an amended Funding Statement declaring this commercial affiliation, as well as a statement regarding the Role of Funders in your study. If the funding organization did not play a role in the study design, data collection and analysis, decision to publish, or preparation of the manuscript and only provided financial support in the form of authors' salaries and/or research materials, please review your statements relating to the author contributions, and ensure you have specifically and accurately indicated the role(s) that these authors had in your study. You can update author roles in the Author Contributions section of the online submission form.

5.2. Please also provide an updated Competing Interests Statement declaring this commercial affiliation along with any other relevant declarations relating to employment, consultancy, patents, products in development, or marketed products, etc.  

6. We note that you have indicated that data from this study are available upon request. PLOS only allows data to be available upon request if there are legal or ethical restrictions on sharing data publicly. For more information on unacceptable data access restrictions, please see http://journals.plos.org/plosone/s/data-availability#loc-unacceptable-data-access-restrictions.

7. We note that you have included the phrase “data not shown” in your manuscript. Unfortunately, this does not meet our data sharing requirements. PLOS does not permit references to inaccessible data. We require that authors provide all relevant data within the paper, Supporting Information files, or in an acceptable, public repository. Please add a citation to support this phrase or upload the data that corresponds with these findings to a stable repository (such as Figshare or Dryad) and provide and URLs, DOIs, or accession numbers that may be used to access these data. Or, if the data are not a core part of the research being presented in your study, we ask that you remove the phrase that refers to these data.

8. Please ensure that you refer to Figure 1 in your text as, if accepted, production will need this reference to link the reader to the figure.

9. Please include captions for *all* your Supporting Information files at the end of your manuscript, and update any in-text citations to match accordingly. Please see our Supporting Information guidelines for more information: http://journals.plos.org/plosone/s/supporting-information.

Reviewers' comments:

Reviewer's Responses to Questions

**Comments to the Author**

1. Is the manuscript technically sound, and do the data support the conclusions?

Reviewer #1: Yes

Reviewer #2: Yes

2. Has the statistical analysis been performed appropriately and rigorously? 

Reviewer #1: Yes

Reviewer #2: Yes

3. Have the authors made all data underlying the findings in their manuscript fully available?

Reviewer #1: No

Reviewer #2: Yes

4. Is the manuscript presented in an intelligible fashion and written in standard English?

Reviewer #1: Yes

Reviewer #2: Yes

5. Review Comments to the Author

Reviewer #1: The present study analyzes the interaction between physical activity and polygenic risk scores for obesity. The authors identify a significant interaction between physical activity and a BMI polygenic risk score (PRS). The effect size of physical activity on BMI among the high-risk PRS was two times greater than the effect size in the low-risk PRS group. In my opinion, this is a methodologically strong paper that contributes to the evidence base in this area. Please see attachment for the complete review.

Reviewer #2: This paper report a well-designed gene x environment interaction study to investigate how genetic liability for obesity interact with physical activity. The results are timely and important to extend our understanding how genetic risk factors could modulate the effect of lifestyle. The manuscript is well structured and clearly written.

However, there are some points that should be considered to improve the manuscript:

1. The authors found that well known risk factors for obesity such as diet, socioeconomic class, mental health and sleep quality were not associated with BMI in their model when PRS was used but the associations were significant when GRS (or perhaps when no genetic variables) was in the model. It would be interesting to know how the authors interpret these findings. Do BMI PRS showed association with these factors? Only associations of cardiometabolic factors were reported in Table S1. To report associations with BMI and BMI PRS of diet, socioeconomic class, mental health and sleep quality would add the quality of the paper. Also, the interpretation should be added to the discussion.

2. It is not clear from the methods how mental health was measured. Please, add further description and reference.

3. Also, it is not understandable what the reported socioeconomic class categories mean, how they were measured. Please, describe it further and provide reference.

4. Please, add further details how genotype QC steps were carried out in the different cohorts. It is especially incomplete for the Genotek dataset.

5. There is no description in the methods section how the LDpred was used to calculate BMI PRS. How was the setting for LDpred defined? Why they selected “fraction of causative variants” 0.3?

6. Naming of the supplementary materials is confusing. Would be better to report them in a uniform way. Furthermore, S1 Table and Table S1 are the same?

7. Do the authors tested whether the other factors (such as diet, socioeconomic class, mental health and sleep quality) interact with BMI PRS, even though they do not have main effect? It is possible to see interaction effect even in the absence of main effect.

8. I miss an overview of past gene x physical activity interaction studies related to obesity or other conditions in the discussion, only FTO was mentioned.

6. PLOS authors have the option to publish the peer review history of their article (what does this mean?). If published, this will include your full peer review and any attached files.

Reviewer #1: No

Reviewer #2: No

---

## [Author Response · Author response to Decision Letter 0]

14 Sep 2021

Thank you to everyone for your exhaustive feedback. Please find our point-by-point rebuttal below.

To the academic editor

 We have changed the style of the title page and headings, corrected minor details in the text, and uploaded the Figures and Supplementary files as separate files following the naming instructions.

 We have not developed any questionnaires as part of this study. We used previously published questionnaires. We described the questions used in detail in the Materials and Methods (lines 129-148), and we have supplied additional references to the papers where the variables were originally published:

• “Smoking status [19], four categories: “smoking daily”, “smoking occasionally”, “never smoking”, and “previously smoking” (N = 5,155).

• Alcohol consumption [20], two categories: “no or moderate alcohol consumption” defined as ≤ 6 units/week for women and ≤ 12 units/week for men; “high alcohol consumption” defined as > 6 units week for women and > 12 units/week for men. 1 unit is equivalent to 12 g of pure alcohol (N = 5,002).

• Diet quality, two categories: “poor diet” defined as 4-8 points on the diet quality score (DQS) system described at [21]; “healthy diet” defined as ≥ 9 points on the DQS score system (N = 5,020).

• Physical activity level [22], two categories: “inactive” defined as self-reported commuting and leisure-time physical activity ≤ 225 min/week; “active” defined as self-reported commuting and leisure-time physical activity > 225 min/week (N = 4,859).

• Mental health [23], two categories: “high” defined as mental health component score (MCS) higher than the 75th percentile within the study population of the same sex; “low” defined as MCS lower than the 75th percentile. MCS has been calculated as described in [23], using the Short Form 12 (SF-12) questionnaire [24] (N = 4,878).

• Quality of sleep, two categories: “good” defined by the answer ‘no’ to the question ‘do you often suffer from insomnia’; “poor” defined by the answer ‘yes’ (N = 5,129).

• Socioeconomic class [20], five categories: “not working, no education”, “not working, ≥ 1 year of education”, “working, no education”, “working, 1-3 years of education”, “working, ≥ 4 years of education”. Education is counted after mandatory school years. The categories were combined from education and employment statuses, reported in [20] (N = 4,807).”

3. PLOS journals require authors to make all data underlying the findings described in their manuscript fully available without restriction unless the data are subject to ethical restrictions or owned by someone other than the authors (https://journals.plos.org/plosone/s/data-availability#loc-acceptable-data-access-restrictions). Therefore, we ask that you please upload underlying data to an appropriate data repository and update your Data Availability Statement accordingly or provide all contact details for where an interested researcher would need to apply to gain access to the relevant data. Please note that it is not acceptable for an author to be the sole named individual responsible for ensuring data access.

 We have initiated uploading our polygenic risk score weight matrix to the appropriate public repository PGSCatalog. However, PGSCatalog is processing the submissions manually, and we are awaiting the approval. In the meantime, we have made the score available via Online supplementary, described in S1 Appendix. We have also shared the underlying code using GitHub, described in S1 Appendix.

4. Thank you for stating the following in the Financial Disclosure section:

"Data collection in the Inter99 study was supported economically by The Danish Medical Research Council (grant nr. 2028-00-0019 and 09-059174), The Danish Centre for Evaluation and Health Technology Assessment (grant nr. 3126-138-1998 and 263-12-1999 and 0-204-03-74), Novo Nordisk, Copenhagen County (grant nr. 9870006), The Danish Heart Foundation (grant nr. 98-2-5-71-22659 and 00-2-9-F4-22872 and 04-10-B201-A309-22171), The Danish Pharmaceutical Association (grant nr. 53-99 and 58-2003), Augustinus foundation (grant nr. 99-1663), Ib Henriksen foundation, and Becket foundation. Novo Nordisk Foundation Center for Basic Metabolic Research is an independent Research Center, based at the University of Copenhagen, Denmark, and partially funded by an unconditional donation from the Novo Nordisk Foundation (www.cbmr.ku.dk) (NNF18CC0034900). Dmitrii Borisevich is receiving funding from NNF Copenhagen Bioscience PhD Programme (NNF17CC0026760).

We note that you received funding from a commercial source: Novo Nordisk

Please know it is PLOS ONE policy for corresponding authors to declare, on behalf of all authors, all potential competing interests for the purposes of transparency. PLOS defines a competing interest as anything that interferes with, or could reasonably be perceived as interfering with, the full and objective presentation, peer review, editorial decision-making, or publication of research or non-research articles submitted to one of the journals. Competing interests can be ompetin or non-financial, professional, or personal. Competing interests can arise in relationship to an organization or another person. Please follow this link to our website for more details on competing interests: http://journals.plos.org/plosone/s/competing-interests

Here is our Competing Interests Statement, including all the required information for the Novo Nordisk and also for Genotek Ltd:

“Novo Nordisk provided unrestricted grants for data collection. No authors were employed by or consulted Novo Nordisk during this study, and no conflict of interest exists in connection to patents, products in development, marketed products, or alike. This funding does not alter our adherence to PLOS ONE policies on sharing data and materials, and does not impose restrictions on sharing of data and/or material. AR and VI are employees of Genotek Ltd. and may own stock/stock options in the company. No other conflict of interest exists in connection to patents, products in development, marketed products, or alike. This funding does not alter our adherence to PLOS ONE policies on sharing data and materials, and does not impose restrictions on sharing of data and/or material. Other authors declare no conflict of interests.”

"I have read the journal's policy and the authors of this manuscript have the following competing interests: Alexander Rakitko and Valery Ilinsky are employees of Genotek Ltd. The rest of the authors declare no competing interests."

We note that one or more of the authors are employed by a commercial company: Genotek Ltd.

5.1. Please provide an amended Funding Statement declaring this commercial affiliation, as well as a statement regarding the Role of Funders in your study. If the funding organization did not play a role in the study design, data collection and analysis, decision to publish, or preparation of the manuscript and only provided financial support in the form of authors' salaries and/or research materials, please review your statements relating to the author contributions, and ensure you have specifically and accurately indicated the role(s) that these authors had in your study. You can update author roles in the Author Contributions section of the online submission form.

The affiliation did not play a role in our study. We have updated the Funding Statement, including the following information:

“Genotek Ltd provided only financial support in the form of AR and VI salaries and the data for the analysis, but did not have any additional role in the study design, data collection and analysis, decision to publish, or preparation of the manuscript. The specific roles of these authors are articulated in the ‘author contributions’ section.”

5.2. Please also provide an updated Competing Interests Statement declaring this commercial affiliation along with any other relevant declarations relating to employment, consultancy, patents, products in development, or marketed products, etc. 

 See the Competing Interests Statement under paragraph 4.

6. We note that you have indicated that data from this study are available upon request. PLOS only allows data to be available upon request if there are legal or ethical restrictions on sharing data publicly. For more information on unacceptable data access restrictions, please see http://journals.plos.org/plosone/s/data-availability#loc-unacceptable-data-access-restrictions.

 The underlying raw data comprises only sensitive data that cannot be made publicly available: genetic and biological data of human participants.

For Inter99 dataset, consent for publication of raw data was not obtained from the individuals, and the dataset could pose a threat to confidentiality. The restriction is imposed by Regional Scientific Ethics Committee (KA 98 155) and the Danish Data Protection Agency. Data access is provided by the Phenomics Platform of the NNF Center for Basic Metabolic Research, https://cbmr.ku.dk/researchfacilities/phenomics/.

For Genotek dataset, the user agreement (available at https://www.genotek.ru) states that disclosure of individual-level genetic information and/or self-reported Information to third parties for research purposes will not occur without explicit consent, and the consent was not obtained from the individuals. Due to the user agreement the individual level cannot be made directly available, and the dataset could pose a threat to confidentiality. Data have to be accessed indirectly via Genotek Ltd, https://www.genotek.ru.

7. We note that you have included the phrase “data not shown” in your manuscript. Unfortunately, this does not meet our data sharing requirements. PLOS does not permit references to inaccessible data. We require that authors provide all relevant data within the paper, Supporting Information files, or in an acceptable, public repository. Please add a citation to support this phrase or upload the data that corresponds with these findings to a stable repository (such as Figshare or Dryad) and provide and URLs, DOIs, or accession numbers that may be used to access these data. Or, if the data are not a core part of the research being presented in your study, we ask that you remove the phrase that refers to these data.

 We had two occurrences, and we have fixed both. We have added Supplementary Table S2, reporting p-values and effect sizes for associations between BMI and the environmental risk factors when adjusted for BMI GRS (line 244). We have removed information about simulations regarding the difference between the variance explained by age and sex alone in the text, as we decided they are not a core part of the research.

8. Please ensure that you refer to Figure 1 in your text as, if accepted, production will need this reference to link the reader to the figure.

We have referred to Figure 1 in the beginning of the Materials and Methods (line 79) with the following text – “The workflow of the analysis is present at Fig 1.”

9. Please include captions for *all* your Supporting Information files at the end of your manuscript, and update any in-text citations to match accordingly. Please see our Supporting Information guidelines for more information: http://journals.plos.org/plosone/s/supporting-information.

 We have added the captions (lines 413-419):

 “Supporting Information

S1 File. Distributions of the BMI-associated phenotypes of individuals, stratified by BMI polygenic risk score (PRS) deciles.

S1 Fig. Performance of models using two groups of genetic risk for interaction with physical activity.

S1 Table. p-values of associations between cardiometabolic traits and BMI or BMI PRS.

S2 Table. Associations of the risk factors with BMI adjusted for BMI GRS.

S1 Appendix. List of online supporting materials.”

 

To the reviewer #1

The present study analyzes the interaction between physical activity and polygenic risk scores for obesity. The authors identify a significant interaction between physical activity and a BMI polygenic risk score (PRS). The effect size of physical activity on BMI among the high-risk PRS was two times greater than the effect size in the low-risk PRS group. In my opinion, this is a methodologically strong paper that contributes to the evidence base in this area. 

Major comments

1. What is the clinical significance of the 78% cut-off? The authors state that this has important clinical implications but this is only shown empirically through the difference in effect size becoming two-fold greater in this group for the interaction. The clinical implications of this are not clear. Are there other cardiometabolic traits that increase significantly (e.g., past a certain threshold or markedly increase disease risk) in the high-risk PRS group? There may be more empirical support needed to justify that this threshold has important clinical implications. 

Thank you for your comment. The significance of the cut-off is utilitarian, in that in clinical genetics practice, it is easier to adopt binary criteria than to use continuous variables with no meaningful scale that a PRS represents. We agree that while we focus on BMI, even though we study a public health topic, we should not make strong statements about clinical implications. Studying how the high-risk PRS group differs from the low-risk group in other cardiometabolic traits would be a very interesting focus for future studies. For the scope of this manuscript, we have removed the statement about the clinical significance from the abstract (line 30):

“Our results … (ii) show a non-linear interaction between BMI genetics and physical activity”.

2. BMI is an imperfect measure of obesity since it does not distinguish fat vs. fat free mass and this warrants mention in the limitations section. 

 Thank you for your comment. We agree with this statement, and we have added the following text (lines 373-378) to the Discussion to address this limitation:

“Third, in this paper we focus on BMI, but there is a piling amount of evidence that other body composition measures are important, when assessing obesity. Fat vs. fat free mass distribution is an important measure [30], capturing differences in body constitution between the people with the same BMI. Another important set of measures are waist-to-hip ratio, waist circumference and similar [31], capturing the fat distribution around the body. These measures are orthogonal to BMI. Future studies of genetics of these parameters using PRS approach could provide additional value to understanding of obesity.”

Minor Comments

Line 45: this section would benefit from describing other sources of missing heritability such as gene-gene interactions.

Thank you for the comment. We agree, and we have listed all major potential sources of missing heritability and provided a reference in the Introduction (lines 43-45):

“This finding represents a marked case of missing heritability. There are many potential reasons for heritability missing from the GWAS findings [7], including non-significant variants with small effect size, rare variants, structural variation, and gene-gene and gene-environment interactions.”

Line 186: is it possible to report the number and descriptive statistics of the individuals that were excluded based on missing data?

 Thank you for the great remark. This was an unfortunate phrasing from our side. 3,415 individuals were available after all exclusions based on age and questionnaire availability. No samples were excluded based on genetic QC, because if a certain sample had low call rate then the corresponding individual was asked to collect the saliva one more time and the analysis started from the beginning. We have removed the filtering description and instead described the dataset in lines 193-194 as:

 “Genotek is an unpublished Russian population-based set consisting of 3,415 unrelated individuals aged between 20 and 60 years old with self-reported measures.”

Line 256: I appreciate the authors’ creativity to remove FTO from the PRS and demonstrate the interaction with physical activity after excluding this variant. Has the interaction with physical activity been tested with any individual SNPs beyond FTO? I appreciate that this may be beyond the scope of this paper but may be worth investigating in future studies if it is a question of interest. 

 Thank you! We have not checked other individual SNPs in our analysis. We agree and strongly believe that it would be interesting to combine together individual SNPs with strong effects and the polygenic background of the rest of the SNPs comprised with a PRS in future studies.

Line 259: I believe “There” should be revised to “These”

 “There” replaces “In the replication dataset” in this context, and it was not a typo. We rephrased it (lines 269-271) as:

 “In this dataset, physically active individuals had 0.97 kg/m2 (p = 9.10 x 10 14) lower BMI than inactive individuals and an additional 0.45 kg/m2 (p = 5.57 x 10-4) lower BMI per one standardized BMI PRS unit.”

Line: 304: it would be informative for the reader to describe the nature of the interaction in the first paragraph of the discussion (e.g., physical activity was associated with a greater decrease in BMI among people with a higher PRS).

 Thank you for the comment, we agree. We have swapped the first and second paragraph of the discussion, to put the paragraph with results and a description of the nature of the interaction on the top of Discussion (lines 305-313):

 “In this study, we have detected a non-linear interaction between BMI genetics and physical activity using BMI PRS. We have constructed a BMI PRS and assessed its performance. The BMI PRS demonstrated a substantial improvement in the explained variance of BMI over the BMI GRS. Using the BMI PRS, we have detected an interaction between the genetic component of BMI and physical activity in a relatively small dataset. This interaction was neither limited to the interaction driven by the FTO locus nor significant when using the BMI GRS for the same analysis. The application of PRS enabled us to identify 22% of individuals with the highest PRS. In this group, self-reported physical activity was associated with a 2-fold higher difference in BMI than in the remaining 78% of study participants. The model with a non-linear two-group division of individuals showed higher R2 than a model with a linear interaction.”

Line 315-316: the flow of this sentence could be improved, perhaps by adding “identify a” after “to”

 We agree, and we have replaced the verb “subset”, which may be confused for a noun, with the proposed “identify” verb.

To the reviewer #2

This paper report a well-designed gene x environment interaction study to investigate how genetic liability for obesity interact with physical activity. The results are timely and important to extend our understanding how genetic risk factors could modulate the effect of lifestyle. The manuscript is well structured and clearly written.

However, there are some points that should be considered to improve the manuscript:

1. The authors found that well known risk factors for obesity such as diet, socioeconomic class, mental health and sleep quality were not associated with BMI in their model when PRS was used but the associations were significant when GRS (or perhaps when no genetic variables) was in the model. It would be interesting to know how the authors interpret these findings.

Also, the interpretation should be added to the discussion.

Thank you for the detailed comment! There seem to be some misunderstanding. We have studied seven risk factors known from the literature. We have found that in our data the four mentioned factors (diet, socioeconomic class, mental health, and sleep quality) did not associate with BMI when PRS was in the model (linear model “BMI ~ 1 + PRS + age + C(sex) + C(risk factor))”). The same risk factors were neither associated with BMI in our data when GRS was in the model (linear model BMI ~ 1 + GRS + age + C(sex) + C(risk factor))), or when no genetic variables were used (linear model BMI ~ 1 + age + C(sex) + C(risk factor))). So, there was no difference between using PRS or GRS. We have added Supplementary Table S2 with the detailed results of the model with GRS to the manuscript.

Do BMI PRS showed association with these factors? 

Since there was no difference between using PRS or GRS, we have not checked whether BMI PRS showed association with the risk factors. To address your question, we have run multinomial logistic regression adjusted for age and sex (general formula “C(risk factor) ~ 1 + PRS + age + C(sex)”). Smoking, alcohol consumption, and mental health were associated significantly with PRS levels at the same cut-off as used for the original analysis (p < 4.17 x 10-3). We think that this might be caused by the fact that a PRS captures variants that are causal for the risk factors since BMI itself is correlated with the risk factors. An increase in the explained variance (calculated using McFadden’s pseudo-R-squared) provided by PRS compared to the models only with sex and age was below 0.3% for each of the traits.

Only associations of cardiometabolic factors were reported in Table S1. To report associations with BMI and BMI PRS of diet, socioeconomic class, mental health and sleep quality would add the quality of the paper. 

Thank you for the comment. We agree that associations between all the risk factors and BMI should be reported, and we have described these associations in Table 1 (see below). The associations between BMI PRS and risk factors were weak, as described in the paragraph above. We think it might distract the readers from the focus of the manuscript, so we have not included these new results in the manuscript. 

“Table 1. Associations of the risk factors with BMI.

Risk factor Category * Effect size P-value

Smoking Never smoker Reference group 

 Previous smoker + 0.06 kg/m2 0.648

 Smoking occasionally + 0.27 kg/m2 0.341

 Smoking daily - 1.06 kg/m2 6.49 x 10-18

Physical activity Active Reference group 

 Inactive + 0.79 kg/m2 8.49 x 10-13

Alcohol consumption No or moderate Reference group 

 High - 0.36 kg/m2 9.00 x 10-4

Diet quality Healthy Reference group 

 Poor - 0.25 kg/m2 0.024

Socioeconomic class Working, ≥ 4 years 

of education Reference group 

 Working, 1-3 years 

of education + 0.00 kg/m2 0.99

 Working, 

no education + 0.21 kg/m2 0.24

 Not working, ≥ 1 year of education + 0.18 kg/m2 0.40

 Not working, 

no education + 0.01 kg/m2 0.98

Mental health High Reference group 

 Low - 0.06 kg/m2 0.62

Quality of sleep Good Reference group 

 Poor + 0.15 kg/m2 0.28

”

2. It is not clear from the methods how mental health was measured. Please, add further description and reference.

3. Also, it is not understandable what the reported socioeconomic class categories mean, how they were measured. Please, describe it further and provide reference.

Thank you for the comments. We agree with both, and we have provided references to all the used questions / measures in the Methods (lines 140-143, 146-151):

“Mental health [23], two categories: “high” defined as mental health component score (MCS) higher than the 75th percentile within the study population of the same sex; “low” defined as MCS lower than the 75th percentile (N = 4,878). MCS has been calculated as described in [23], using the Short Form 12 (SF-12) questionnaire [24].

Socioeconomic class [20], five categories: “not working, no education”, “not working, ≥ 1 year of education”, “working, no education”, “working, 1-3 years of education”, “working, ≥ 4 years of education”. Education is counted after mandatory school years. The categories were combined from education and employment statuses, reported in [20] (N = 4,807).

All risk factors were measured by self-report questionnaire in the Inter99 cohort.”

4. Please, add further details how genotype QC steps were carried out in the different cohorts. It is especially incomplete for the Genotek dataset.

 Thank you for your comment, we have described the details of genotyping QC tools and parameters:

Inter99 dataset, lines 184-188:

“Genotypes were called using the Genotyping module (version 1.9.4) of GenomeStudio software (version 2011.1; Illumina). Only individuals having a call rate ≥98% were included. Monomorphic SNPs and SNPs with Hardy-Weinberg expectation p-value < 10-5 were excluded. Genotypes were imputed using the Michigan imputation server with the HRC1.1 [26] reference panel.”

Genotek dataset, lines 195-198:

“All individuals were genotyped using Illumina Infinium Global Screening Array (v1.0 / v2.0). SNPs with call rate <0.9, the calls on Y chromosome for women and the heterozygous calls on X chromosome for men were removed. Genotypes were imputed using BEAGLE 5.1 with the HRC reference panel. After imputation, variants with MAF below 1% or DR2 below 0.7 were excluded.”

5. There is no description in the methods section how the LDpred was used to calculate BMI PRS. 

How was the setting for LDpred defined? Why they selected “fraction of causative variants” 0.3?

Thank you for your comment. We agree that the description was not clear enough. We have defined the “fraction of causative variants” parameter at 0.3 by screening eleven different potential fractions from evenly spaced logarithmic scale in the training subset. We have added a more detailed LDpred protocol to the Methods and described the screening process in more detail (lines 92-102):

“PRS was generated using LDpred tool (v.1.0.6) [9] and its standard workflow. Briefly, a Danish population-based cohort Inter99 [18] (N = 6,179) was used. The cohort was randomly split into training (N = 1,000) and validation (N = 5,179) subsets. SNPs available in the input data, comprising subsets and the summary statistics of the BMI GWAS, were aligned using LDpred coord command. Eleven BMI PRS were generated for the different parameter f, representing what LDpred calls the “fractions of causative variants”. The following parameters f were used to cover different orders of magnitude: 1.0, 0.3, 0.1, 0.03, 0.01, 0.003, 0.001, 0.0003, 0.0001, 3 x 10-5, 1 x 10-5. The scores were generated using LDpred gibbs and their performance was assessed in the training subset only using R2 provided by LDpred score command. The best performing score was selected. The score values were calculated for the samples in the validation subset using LDpred score command. These values and this subset were used for the subsequent analyses.”

6. Naming of the supplementary materials is confusing. Would be better to report them in a uniform way. 

We have adjusted our naming. Please, let us know if there is still any confusing naming.

Furthermore, S1 Table and Table S1 are the same?

 Thanks for catching this, they are indeed the same. We have replaced “S1 Table” with “Table S1”.

7. Do the authors tested whether the other factors (such as diet, socioeconomic class, mental health and sleep quality) interact with BMI PRS, even though they do not have main effect? It is possible to see interaction effect even in the absence of main effect.

Thank you for your comment. The way we described it made it look as if we only checked the three significantly associated risk factors, but we have actually checked all seven factors. We have rephrased this block in the Results section (lines 251-254):

“To analyze if the presence of the risk factor alters the effect of BMI PRS on BMI, we examined potential interactions between the BMI PRS and all seven risk factors (linear regression BMI ~ 1 + PRS + age + C(sex) + C(risk factor) + PRS:C(risk factor)). We found a significant interaction (twelve tests performed, p < 4.17 x 10 3) only between the BMI PRS and physical activity.”

8. I miss an overview of past gene x physical activity interaction studies related to obesity or other conditions in the discussion, only FTO was mentioned.

We have updated the Introduction to cover both studies discovering individual SNP-environment interactions, including FTO and CDH12 (rs986732), and studies on interactions visible via aggregated genetic background using risk scores in the following way (lines 53-63):

 “Another source of the unexplained variation of BMI stems from gene-environment interactions [11,12]. An interaction occurs when the biological effect of a genetic variant depends on a risk factor, such as an environmental stimulus or a lifestyle factor [13,14]. For example, physical activity attenuates the effect of a common SNP rs9939609 within the FTO locus on BMI [15,16], the strongest common SNP known to associate with BMI. However, detection of the interactions driven by individual SNPs is challenging. A meta-analysis of 200,452 adults [12] reported only one additional SNP, rs986732, on top of the known FTO locus. Attempts to aggregate genetic background using GRS to increase power to detect gene-environment interactions have been made. Recently, a large study in up to ~360,000 unrelated participants from UK Biobank identified several risk factors – alcohol intake, physical inactivity, socioeconomic status, mental health, and sleeping patterns – that influenced the effect of a BMI GRS comprised of 94 BMI-associated genetic variants on BMI [17].”

---

## [Editor Report · Decision Letter 1]

5 Oct 2021

Non-linear interaction between physical activity and polygenic risk score of body mass index in Danish and Russian populations

PONE-D-21-17351R1

Dear Dr. Borisevich,

We’re pleased to inform you that your manuscript has been judged scientifically suitable for publication and will be formally accepted for publication once it meets all outstanding technical requirements.

Kind regards,

David Meyre

Academic Editor

PLOS ONE
---

## [Editor Report · Acceptance letter]

8 Oct 2021

PONE-D-21-17351R1 

Non-linear interaction between physical activity and polygenic risk score of body mass index in Danish and Russian populations 

Dear Dr. Borisevich:

I'm pleased to inform you that your manuscript has been deemed suitable for publication in PLOS ONE. Congratulations! Your manuscript is now with our production department. 

Kind regards, 

on behalf of

Dr. David Meyre 

Academic Editor

PLOS ONE